# An Interprofessional E-Learning Resource to Prepare Students for Clinical Practice in the Operating Room—A Mixed Method Study from the Students’ Perspective

**DOI:** 10.3390/healthcare9081028

**Published:** 2021-08-11

**Authors:** Ann-Mari Fagerdahl, Eva Torbjörnsson, Anders Sondén

**Affiliations:** 1Department of Clinical Science and Education, Karolinska Institutet, 118 83 Stockholm, Sweden; eva.torbjornsson@ki.se (E.T.); anders.sonden@ki.se (A.S.); 2Wound Centre, Södersjukhuset, 118 83 Stockholm, Sweden; 3Department of Surgery, Södersjukhuset, Sjukhusbacken 10, 118 83 Stockholm, Sweden

**Keywords:** clinical learning environment, e-learning, operating room, student preparedness

## Abstract

The operating room is a challenging learning environment for many students. Preparedness for practice is important as perceived stress and the fear of making mistakes are known to hamper learning. The aim was to evaluate students’ perspectives of an e-learning resource for achieving preparedness. A mixed methods design was used. Students (*n* = 52) from three educational nursing and medical programs were included. A questionnaire was used to explore demographics, student use of the e-learning resource, and how the learning activities had helped them prepare for their clinical placement. Five focus group interviews were conducted as a complement. Most students (79%) stated that the resource prepared them for their clinical placement and helped them to feel more relaxed when attending to the operating room. In total, 93% of the students recommended other students to use the e-learning resource prior to a clinical placement in the operating room. Activities containing films focusing on practical procedures were rated as the most useful. We conclude that an e-learning resource seems to increase students’ perceived preparedness for their clinical practice in the operating room. The development of e-learning resources has its challenges, and we recommend student involvement to evaluate the content.

## 1. Introduction

The operating room (OR) environment is challenging for students in relation to achieving their learning objectives. Feelings of anxiety, humiliation, and other emotional obstacles for effective learning have been described by both medical and nursing students [1,2]. Some of these emotional barriers can be reduced if the students are well-prepared before their clinical practice [3,4]. Preparedness can be divided into a general part and a specific part. The general part should consist of information about the OR setting, etiquette, and the professional roles of the staff, in combination with workshops on practical skills. The specific part is the information needed on day-to-day basis, i.e., which supervisor the OR student should follow [1,2].

Methods for delivering general introductory sessions to students have been described by several authors, but there is weak evidence as to which arrangement is the most effective [2,5]. It has been concluded, however, that the introductory sessions should have an interprofessional perspective, as interprofessional teamwork is essential for creating a safe surgical environment for patients [6]. However, interprofessional learning (IPL) activities pose logistical and scheduling challenges [7]. One way to overcome these timetabling and geographic barriers is e-learning [8]. Another advantage of e-learning is that it is well-suited for learning practical skills within the perioperative setting, due to the possibility to incorporate multimedia [8,9]. 

For many years, the medical students and the OR nurses in our OR department at Södersjukhuset, Karolinska Institutet have attended a pre-theatre workshop on surgical hand preparation and sterile gloving technique before entering the OR. The workshop contains a lecture, followed by practical training. The general nurses and anesthetic nurses have only a 15 min lecture about guidelines for clothes and aseptic techniques. A survey aimed at the medical students in 2016 showed that the students perceived that the general introduction was too sparse; moreover, the practical workshop was too short, and it lacked an interprofessional approach. Therefore, an interprofessional faculty at our institution created a complementary e-learning resource, defined as a software-based resource distributed online with the aim to enhance knowledge and performance [9] for all students attending the OR [10].

The aim of the e-learning resource was to better prepare the different student categories (nurses, OR nurses, anesthetic nurses, and medical students) and to reduce emotional barriers, hence creating a better foundation for learning. The focus of the learning outcomes in the e-learning resource was set on skills and interprofessional collaboration.

In 2018, we performed a pilot study evaluating the e-learning resource. It was concluded that it was valuable to the students, but it was difficult to draw conclusions on why and how it was valuable due to lack of qualitative data. Moreover, only medical students participated in the evaluation, so no conclusions could be drawn for the other student categories. There was also a lack of knowledge regarding ideas for improvement of the e-learning resource.

The aim of this study was thus to explore the perspectives of all student categories using the new e-learning resource, with a focus on preparedness for practice.

## 2. Materials and Methods

### 2.1. Design

An explanatory sequential mixed methods design was used, i.e., data was collected in two consecutive phases: first the quantitative data and then the qualitative data. Thereafter, the data was merged to achieve methodological integration [11,12]. Questionnaires were used to gather qualitative data. Focus group (FG) interviews were used to deepen the knowledge from the questionnaires and to obtain suggestions and ideas for improvement of the e-learning resource. This specific qualitative data collection method was chosen for its ability to help participants to explore and explain their perceptions further in interaction with others [13]. 

### 2.2. The E-Learning Resource

The e-learning resource used in this project was a package of online learning materials using Articulate Storyline^®^ (Articulate Global, New York, NY, USA) and consisted of pre-recorded lectures and video demonstrations of skills which could be accessed on different digital devices such as computers or mobile phones. The software used to produce the learning material were PowerPoint^®^ (Microsoft Corporation, Redmond, WA, USA) and Screencast-omatic ^®^ (UserVoice, San Francisco, CA, USA), while the films were recorded using a regular camcorder with a microphone.

The resource was based on seven interprofessional learning outcomes, each one forming the base for a learning activity in the online program. The majority of the learning outcomes were considered generic, except “surgical hand preparation” and “gowning procedure” that were directed to the OR nurses and medical students exclusively (Table 1). 

Four of the learning activities were followed by a formative assessment in order to give immediate feedback to students.

### 2.3. Participants

Students from three educational programs were included in the study: 4th year medical students (*n* = 24), 3rd year nursing students (*n* = 12), and 1st year perioperative specialist nursing students, specializing in either OR nursing or Anesthesiology nursing (*n* = 16). The medical and nursing students all had their clinical placement at the OR ward in the same hospital, while the perioperative nursing students did their clinical placement in two different hospitals in Stockholm connected to the university. The students received written information regarding the study in their ordinary online learning management system (Ping-Pong AB, Stockholm, Sweden) and verbal information in their course introduction at campus. An email to all eligible students was sent with information on the e-learning resource and the study, together with a link to the e-learning resource on their study platform. All nursing students had a link to the evaluation questionnaire on their study platform. The medical students were given the questionnaire on paper during their examination week at the end of the semester. All students received information regarding the focus groups (FG), and the students who were willing to participate were invited to contact the researcher by mail. In total, 52 students (33 women and 19 men) out of 117 enrolled in the studied programs answered the questionnaires, giving a response rate of 44%. Out of them, 65% had used the e-learning resource prior to their clinical placement (Table 2).

### 2.4. Data Collection

#### 2.4.1. Questionnaire

The questionnaire was developed by the research group and was based on the questionnaire used in the pilot study by Torbjornsson et al. [10]. To address face and content validity, the questionnaire was discussed within the expert group and modified by adding further questions and using another scale for the answers (a 5-level Likert scale instead of a 4-level) [14]. None of the students asked questions about the questionnaire that suggested that they had difficulties to understand it. 

The questionnaire consisted of 16 questions: 4 were demographic, 3 contained information regarding the use of the e-learning resource, and 9 were questions where the students rated how well the different learning activities had helped them prepare for their clinical placement in the OR (on a 5-level Likert scale: very little; little; some; large; very large). There was also one open-ended question where the students could give improvement suggestions on the resource (Appendix A). 

#### 2.4.2. Focus Group Interviews

The FG interviews focused on evaluating the e-learning resource and the students’ perceptions regarding if and how it helped them to prepare for their clinical practice. The students were divided into groups based on their profession. The aim was to create homogeneity in the groups and avoid any form of hierarchy that may inhibit an open atmosphere enabling everyone to feel confident to speak out [13]. 

The FG were attended by a moderator and conducted by the first and second author. The FG interviews lasted 22–45 min and were documented by note-taking from the moderator. A semi-structured interview guide was used, and probing questions were used to further enable the participants to elaborate. Five FG interviews were conducted with a total of 17 students (2–5 students/FG): medical students (*n* = 9), nursing students (*n* = 2), and perioperative specialist nursing students (*n* = 6). All of the focus groups contained participants of the same educational program.

### 2.5. Data Analysis

#### 2.5.1. Questionnaire—Quantitative Data

The quantitative data analysis was performed on the questionnaires from the students with descriptive statistics [15]. Continuous variables are presented with mean and standard deviation (SD) and categorical variables as *n* (percent). No comparative analyses between the different student categories were made. The quantitative analyses were performed with IBM SPSS statistics version 23.0 (IBM Inc., Chicago, IL, USA). 

#### 2.5.2. Focus Group Interviews—Qualitative Data

The notes from the FG were read and reread to identify patterns and tendencies. The text units were condensed into meaning units, labelled with a code, and sorted into different categories based on the focus areas of the questionnaire. The focus of the qualitative analysis was to extend and to deepen the knowledge from the questionnaires. 

The analysis was performed by two members (AMF and ET) of the research group, and the result was discussed until consensus was reached. Directed content analysis using a deductive approach, based on the different areas of the questionnaire, was used [16].

### 2.6. Ethical Considerations

The study was performed in accordance with good clinical practice and research as per the Helsinki Declaration [17]. Before any data collection began, the students were informed that the participation in this study was voluntary with the purpose of a scientific analysis and publication. They were also informed that their participation in no way would affect their grades, that they could cease participation at any time, and that the collected data would be completely discarded if they were to withdraw from the study. Completing and returning the questionnaire implied their consent to participate. To ensure confidentiality, one of the three authors did the initial analysis of the questionnaires and FG interviews and matched the participants’ data using numeric codes.

## 3. Results

The demographics of the study population is shown in Table 1. The majority of the students (79%) stated that the e-learning resource had prepared them for their clinical placement in the OR, and the medical students rated the e-learning resource as the least useful. However, three quarters of the medical students still rated the e-learning resource as useful to some extent and none rated it as not useful at all (Figure 1).

In total, 93% of the students recommended other students to use the e-learning resource prior to a clinical placement at the OR ward. The differences between student groups are shown in Figure 2. 

### 3.1. The E-Learning Resource in Preparation for Clinical Placement at the OR

The students were asked to what extent the different learning activities of the e-learning resource had helped them prepare for their placement at the OR. Eighty percent of the students perceived that all the different learning activities, at least to some extent, had helped them prepare for their clinical placement (Figure 3). 

The FG interviews revealed that many of the students felt different levels of anxiety prior to their clinical placement at the OR. The students explained that the main cause of this anxiety was the feeling of being in an unfamiliar environment and a sense of being out of place. These feelings gave rise to insecurity and stress and made them more sensitive and vulnerable to what people said or to events that occurred. Students from all programs thought that the e-learning resource was a way to reduce the perceived stress and had prepared them for the OR placement:
*“You want to be as well prepared as possible when you arrive. Although you may not be able to practice so much practically wearing gloves and such things, you want to be able to see… because then it is good because then you can see it over and over again… and you become a little more confident when you come out if you have seen it…”**(Perioperative nurse)*

In the FG interviews, all three student categories commented that the e-learning resource should be a mandatory learning activity for all student groups prior to clinical placement at the OR. 

The nursing students stated that the e-learning resource saved time for them: by being better prepared, they could use their time in the OR more efficiently. Medical students, having a two-hour OR preparation workshop prior to attending their clinical placement, thought the workshop per se prepared them well, but that the e-learning resource was a good complement.

### 3.2. The Students’ Perception of the Content in the Learning Resource

In total, 52% of the students perceived, to a large or very large extent, that the e-learning resource contained all the elements needed to prepare them for their clinical placement. The perioperative nursing students rated the content highest (67%) while the medical students expressed a need for additional information. 

The three learning activities containing films and focusing on practical procedures (‘gloving technique’, ‘surgical hand preparation’, and ‘gowning procedure’) were rated most useful (Figure 3). All of the medical students and a majority of the other student categories perceived that the latter two activities, to a very large extent, had prepared them for the OR. 

The least valuable activities according to the students were ‘OR-design’ and ‘radiation safety’. ‘Radiation safety’ got the lowest ratings of all activities, particularly by the medical and perioperative nursing students (Figure 4). 

The FG interviews revealed that the medical students and perioperative nursing students wanted more information directed to their special needs in their specific profession. The medical students expressed a wish for films that could give them a general overview of the workflow and patient process in the OR, together with instructions on what happens when something goes wrong—for example, what to do if they are unsterile during surgery. They thought this would ease the stress and fear of doing something wrong.
*“Another suggestion is to also inform about what happens when something goes wrong, for example when we get unsterile or similar. So that when and if it happens it will not be so huge but you know what to do if it happens and how to handle this. Takes a little bit of the stress boost…”**(Medical student)*

The medical students also requested films containing specific surgical specialties depending on what kind of surgery they were doing in their clinical placement. The perioperative nurses mentioned elements like positioning on the OR bed and instrument knowledge. The nursing students proposed that the e-learning resource should be divided into two parts: part 1 with basic interprofessional content, which would be mandatory for all students, and part 2 with optional, more profession-oriented content.

### 3.3. Design and Layout of the E-Learning Resource

The majority of the students stated that they had used the e-learning resource on their computer and not on their cell phone. The reason was that the resource was not adequately adapted to the cell phone format. The students thought that they would have used the e-learning resource more often if it was better adapted to the cell phone. This was particularly important for the activities containing films.

The perioperative nursing students and the nursing students experienced the resource as being too messy and lacking a well-defined flow. This was a major obstacle when conducting the activity.
*“It is confusing the whole arrangement I think... you do not know which (learning activity) one is inside and so you click back, and you end up somewhere else... it is difficult to remember...”**(Perioperative nursing student)*

The medical students agreed that there could be a more structured arrangement; however, they did not consider this to be a major issue. 

All student groups commented that the technical form of the learning activity had some drawbacks. The most important part identified by all the focus groups was that the students wanted to be able to see which activities they had performed, which they had left, and finally when they had succeeded with the entire learning activity. Moreover, they wanted to see how long the films were, how much time had passed, and how much was left. One suggestion from the medical students was that the activities could change color when they had been performed. Students from all categories also wished for the possibility to pause and to rewind if they needed to repeat something, without the need to restart a module in the e-leaning resource.

The students all agreed that the maximum amount of time for this kind of learning resource should not be more than 30 min and that the films should not exceed five minutes each. 

### 3.4. Interprofessional Perspective

The learning activity ‘Professions’ was rated differently by the student groups. Common for all student groups was that they considered the interprofessional knowledge in the e-learning resource an important feature, that it was important to learn about each other’s responsibilities at the OR, and that this was essential for a successful interprofessional collaboration in the future. However, the FG interviews identified several requests for improvement regarding the interprofessional approach. Students wanted deeper knowledge regarding the task of the different professions working at the OR ward and not only, as earlier described, a film focusing on the patient process or journey throughout the surgical procedure at the OR. They also proposed an additional film following the different professions in their daily work.
*“We are lacking an overall picture of what is happening at the operating room, a description of the flow. To be able to prepare even better. A kind of “patient journey” through the flow to the surgery department and also a “staff journey” to gain an increased understanding of other professions in the surgery. That is generally lacking in teaching in general.”**(Medical student)*

## 4. Discussion

Every semester, the OR receives students from different education healthcare programs. Many of them, regardless of student category, perceive the learning environment at the OR as extremely stressful [1,2]—something that is known to hamper their learning [18]. This study shows that an e-learning resource based on seven interprofessional learning outcomes enhanced the different student categories’ perceived preparedness for their clinical placement. It did so by making the environment less unfamiliar by explaining the expected role of the student as well as the role of the other professions at the OR and how they interact. The e-learning resource also gave students the possibility to learn specific skills that are known to induce stress when performed in the real environment [19]. These learning activities could be seen “over and over again” and be repeated just before practice. 

The students rated the activities that contained film and focused on practical skills as most valuable. This is consistent with previous findings that skills training such as sterility and operating room etiquettes have been seen as particularly important [2]. Radiation safety got the lowest rating. The rationale for that part could be discussed. It might have been better to name the activity ‘safety in the OR’ and include, for example, laser safety and how to manage surgical smoke. The lack of student involvement is probably one of the reasons that we missed that in the design. One can also argue that knowledge regarding radiation may be quite abstract for the students and that it does not seem as important as the practical skills [18]. Overall, the findings from the FG interviews conclude that it is important to have student involvement in the design phase of an e-learning resource. For example, there was a request from both the medical students and the perioperative nurses to have more films that contained information regarding specific skills, such as instrument knowledge and surgical skills. 

The format is of importance when developing online learning material. The students in this study demanded easy access, preferably in cell phones, with short films and an easy way to find the different contents of the learning activities. Since digital use in society has exploded in recent years and the generation of today’s students have been using digital devices their entire life, they would be a useful resource when creating different online activities. Haraldseid, Friberg, and Aase [20] concluded that active student involvement in the development of technological learning material for clinical skills training could enhance the knowledge of the most important learning needs of the students. It could also make the learning activities more effective and attractive [20]. 

Interprofessional collaboration is known as a major stressor for students attending the OR. Students fear to be despised by the surgeon or nurse when doing something wrong or for simply being in the room [2]. The students expressed the need for knowledge regarding actions when doing something wrong, to be better prepared for such situations. In the FG interviews, the students expressed that they had identified the importance of interprofessional collaboration by using the e-learning resource, and they requested a deeper knowledge of the functions of the different professions, despite the resource not being an IPL resource by definition [21]. The students expressed the IPL ground of learning with and particularly about each other as an area that should be expanded, since they experienced this as important for their psychological preparation for clinical placement at the OR.

Just-in-time teaching (JiTT) is a learning model shown in research to enhance student motivation and to give the students a sense of control. JiTT is defined as a method where the students prepare just before the lesson and lesson time, focusing on specific questions that they experience as difficult and demanding [22]. This e-learning resource may be seen as a way of using the JiTT method, since the students can go through the activities just prior to the clinical task and be better prepared. The main advantage is to be able to repeat the specific element as many times as needed for the student in an easy-access way (on their cell phone, for instance). This creates the possibility for students to tailor their learning to meet their own specific individual learning needs [9,23], which is particularly important given that we address such a broad spectrum of different students. Furthermore, in a stressful environment like the OR, the stress and anxiety of students may inhibit learning and prolong the learning curve [2]. To prepare students by using e-learning in the practical procedures, they may feel less stress when arriving at the OR, and the threshold for learning can be lowered. 

The development of an e-learning resource, such as the one that we have described, could be a useful learning method for student groups in other contexts where practical skills and interprofessional collaboration is important.

### Limitations

It may be argued that a limitation of this study is that we did not have a control group and did not perform a comparative study to assess the effectiveness of the e-learning resource. The aim of the study was, however, not to measure student preparedness or specific knowledge or skills, but to evaluate and describe the students´ perceptions and to explore the value of the resource from the students´ perspective. 

It is recommended that the ideal size of a focus group is six to ten people [24]. In our study, we had a convenience sample which was based on the voluntariness and interest of the students, and we unfortunately did not manage to receive further participants. However, Cote-Arsenault and Morrison-Beedy [25] emphasize that the aim of the study together with developmental levels of the participants is more important than a set number. Since we included students with several years of university studies and we chose to have the FGs separated for the different categories, we believe that the low number of participants in the groups did not inhibit the creativity and data received, nor did we feel that the participants felt pressured to speak, which is described as being a risk in low-numbered FGs [25] 

To only rely on notetaking during the FG interviews and not audiotapes is a limitation. However, as the moderators had high knowledge regarding the setting and the appearance of the e-learning resource, it was not perceived as a problem. The low response rate in the quantitative part of the study can seem to be a problem for the validity (44%). There is a risk of nonresponse bias; however, it is tempting to believe that it does not have the same impact on the result as it may have when it comes to sensitive data such as aspects on quality of life [26]. 

It also needs to be mentioned that the questionnaire was not evaluated with a psychometric test. However, we believe the fact that the questionnaire was evaluated in the expert group, as well as in the pilot study, increases the validity. Further, the use of FG gave a deeper knowledge regarding the e-learning resource.

## 5. Conclusions

We conclude that an e-learning resource seems to increase students’ perceived preparedness for their clinical practice at the OR. The students stated that they felt more relaxed when attending the OR, which may, according to the literature, lead to a better learning environment and improved learning. The development of e-learning resources has its challenges, and we recommend student involvement to evaluate the content of the learning activities as well as to prevent technical issues.

## Figures and Tables

**Figure 1 healthcare-09-01028-f001:**
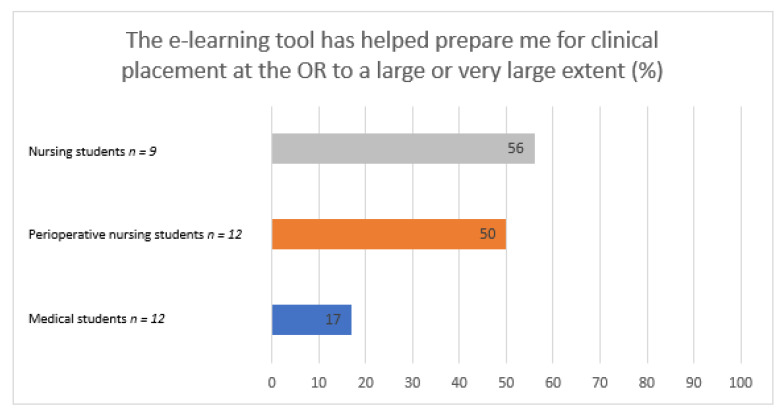
Feelings of preparedness for clinical placement at the OR.

**Figure 2 healthcare-09-01028-f002:**
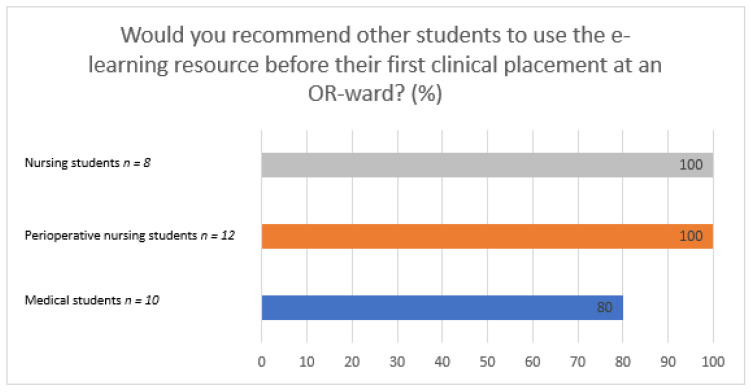
Students’ recommendation of the resource to other students.

**Figure 3 healthcare-09-01028-f003:**
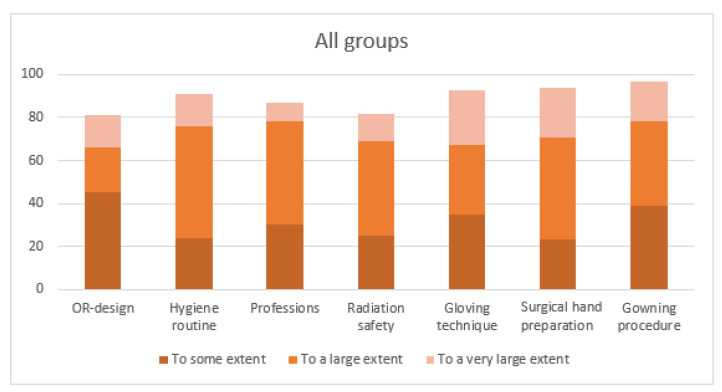
Preparedness for the clinical placement at the OR regarding the different learning activities.

**Figure 4 healthcare-09-01028-f004:**
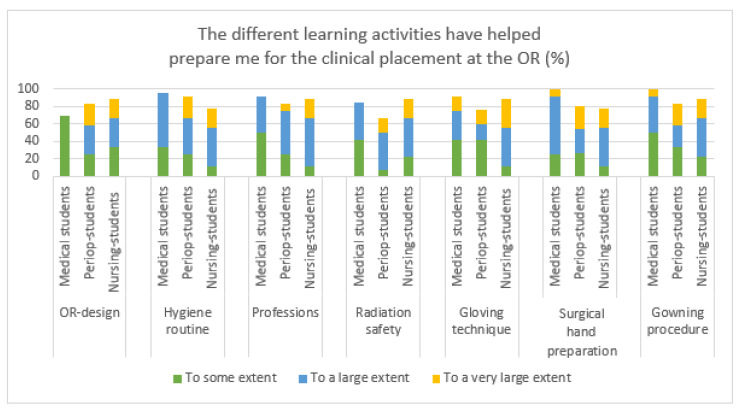
Preparedness for clinical placement at the OR regarding the different learning activities from an all-professions perspective.

**Table 1 healthcare-09-01028-t001:** Learning outcome and learning activities of the e-learning resource (Torbjornsson et al., 2018).

Learning Outcome	Learning Activity	Format	Running Time (min)
The student shall understand the structure of an operation ward	OR design	Recorded audio lecture	5.04
The student shall understand the different professions working at the OR and their responsibilities	Professions	Recorded audio lecture	4.24
The student shall have knowledge regarding the hygiene routine at the OR	Hygiene Routine	Recorded audio lecture	0.45
The student shall be able to describe the radiation safety at OR	Radiation Safety	Recorded audio lecture	2.34
The student shall be able to perform a sterile gloving technique	Gloving technique	Recorded audio movie	1.40
The student shall be able to perform a perioperative surgical hand preparation	Surgical hand preparation	Recorded audio movie	4.01

**Table 2 healthcare-09-01028-t002:** Demographics of the study population.

	All Students *n* = 52	4th Year Medical Students *n* = 24	1st Year Perioperative Nursing Students *n* = 16	3rd Year Nursing Students *n* = 12
**Age mean (range)**	34.0 (21–55)	29.5 (22–47)	38.4 (26–55)	34.9 (21–52)
**Gender (%)**				
Male	33 (63)	13 (54)	4 (25)	2 (17)
Female	19 (37)	11 (46)	12 (75)	10 (83)
**Previous experience of OR (%)**				
Yes	31 (60)	12 (50)	13	6 (50)
No	19 (37)	10 (42)	3	6 (50)
Missing	2 (3)	2 (8)		0
**Had used the e-resource (%)**				
Yes	34 (65)	13 (54)	12 (75)	9 (75)
No	18 (36)	11 (46)	4 (25)	3 (25)
Missing	0	0	0	0

## Data Availability

The data presented in this study are available on request from the corresponding author.

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
