# Peer review of "An Interprofessional E-Learning Resource to Prepare Students for Clinical Practice in the Operating Room—A Mixed Method Study from the Students’ Perspective"

_healthcare, 2021, doi:10.3390/healthcare9081028_

Round 1
Reviewer 1 Report
Dear authors,
Thank you for this study, which is interesting and important. It has however several aspects which should be corrected and improved.
Introduction: page 2, line: 47 - you are refering to your institution, which is however not directly indicated. In my opinion, it would be better to address this issue in more formal way. Could you for example, describe the curriculum in the scope of OR preparation (showing nursing and medical studies)? Is there any content included regarding OR? If yes, how many hours, what scope of material, at what stage of nursing/medical studies? Than, your university initiative (e-learning resource) can be shown in more detail. In this way readers can see the justification for your initiative and context of education regarding OR
extract from page 3, line 116 - regarding response rate, I suggest to move it to results section, for the begining; or to the section describing participants
Regarding FG - could you clarify whether FG interviews groups included mixed students (from different faculties)? What was the number of students in one group?
The qualitative analysis method should be described in more details
Did you receive any approval from ethical comittee for conducting this study? Please, could you state whether researcher(s) were at the same time teachers of students-participants in the study? If so, how did you manage with avoiding any preasure that could have been experienced by students? What kind of consent to take part in the study was given by students (oral/written)?
Results, especially coming from qualitative data, should be strenghten with quotations. This material is described in non systematic way and it should be improved
Conclusions. I think, you should consider more international context to show how students benefit from earlier preparation to OR placement. To do so, better description of e-learning resource developped by your institution is needed in the Introduction or material/method section, in order to show why this may be helpful not only in Sweden.
Author Response
Thank you for your valuable comments and the opportunity to improve our manuscript.
Introduction: page 2, line: 47 - you are refering to your institution, which is however not directly indicated. In my opinion, it would be better to address this issue in more formal way. Could you for example, describe the curriculum in the scope of OR preparation (showing nursing and medical studies)? Is there any content included regarding OR? If yes, how many hours, what scope of material, at what stage of nursing/medical studies? Than, your university initiative (e-learning resource) can be shown in more detail. In this way readers can see the justification for your initiative and context of education regarding OR
Answer: Thank you for your comment and we do see your point. The curriculum of the students attending the OR differs to a large extent, between programs and in different countries and it would be almost impossible to give an overview of it. However, we have tried to expand the information how we prepare our different student categories at our University.
At the same time, while creating the e -learning resource it was shown that the learning outcomes, in the scope of preparing the different student for the OR, were common for most students and we believe this can be the case in most countries. These learning outcomes are spelled out in Table 1 and we hope the readers can use them to judge whether our results can be transferred to their context.
Information regarding the stage of nursing/medical studies is provided in the method section. (Participants).
- extract from page 3, line 116 - regarding response rate, I suggest to move it to results section, for the beginning; or to the section describing participants
Answer: Thank you for your suggestion. We have moved the information to the section of participants.
- Regarding FG - could you clarify whether FG interviews groups included mixed students (from different faculties)? What was the number of students in one group?
Answer: The FG were homogenous containing only participants from the same study professions and the number of students in the groups ranged from two to five. This information has been added in the manuscript.
- The qualitative analysis method should be described in more details
Answer: Thank you for your comment. We have added some clarifying information in the method section under analysis.
- Did you receive any approval from ethical committee for conducting this study? Please, could you state whether researcher(s) were at the same time teachers of students-participants in the study? If so, how did you manage with avoiding any pressure that could have been experienced by students? What kind of consent to take part in the study was given by students (oral/written)?
Answer: we did not apply for approval from an ethical committee since this is not needed according to Swedish law when the study was not aiming on health or other sensitive personal data (the Swedish Ethics Review Act (2003:460)). Permission was received from the head of each study program at the university. The researchers performing the data collection in practice were not involved in the participants educational program and we believe therefore not being any pressure or other impact on the participants, thus not mentioned in the manuscript. Completing and return of the questionnaire, either in written form (medical students) or digitally (nursing students) implied consent to participate as described in the manuscript.
- Results, especially coming from qualitative data, should be strenghten with quotations. This material is described in non systematic way and it should be improved
Answer: additional quotations to strengthen the result have been added. We have tried to start each part in the result section with the quantitative result from the questionnaires followed by deeper explanations from the qualitative analysis of each area in focus. Some revisions have been made in the result section of “3.3. Design and layout of the e-learning resource” for at clearer systematic presentation of the result.
- Conclusions. I think, you should consider more international context to show how students benefit from earlier preparation to OR placement. To do so, better description of e-learning resource developed by your institution is needed in the Introduction or material/method section, in order to show why this may be helpful not only in Sweden.
Answer: We have revised the conclusion and we have extended the information regarding the e-learning resource in the method section to address this issue.
Reviewer 2 Report
A very professionally written paper which is of great interest to those working in the operating units of the hospitals.
I have the following comments:
- The authors state that: “Baseline differences in categorical variables were evaluated with Pearson Chi Square test” The results of these evaluations is missing in the paper.
- The conclusion is not supported by any statistical data.
- The e-learning resource could be referred to, linked to, in a way enabling the reader to collect information.
- The questionnaire used could be included in a supplementary material.
Author Response
Thank you for your valuable comments and the opportunity to improve our manuscript.
- The authors state that: “Baseline differences in categorical variables were evaluated with Pearson Chi Square test” The results of these evaluations is missing in the paper.
Answer: Thank you for pointing out this mistake to us. The sentence has been removed from the text since no comparative analysis were made.
- The conclusion is not supported by any statistical data.
Answer: We have re-written the conclusion to better relate to the qualitative result and to be clearer that it does not result from statistical calculations.
- The e-learning resource could be referred to, linked to, in a way enabling the reader to collect information.
Answer: Some of the information regarding the e-learning resource have been moved to the section “2.2. The e-learning resource” and additional information regarding the resource has been added. Unfortunately we will not be able to link to the resource since it is not able to access publicly.
4.The questionnaire used could be included in a supplementary material.
Answer: Thank you for the suggestion. A translated version of the questionnaire has been included in a supplementary material.
Round 2
Reviewer 1 Report
Corrected manuscript is acceptable now.